# Effects of five teaching methods in clinical nursing teaching: A protocol for systematic review and network meta-analysis

**Jinhui Ni[1]⊚, Pei Wu[2]⊚, Xinlin Huang[1‡], Fangfang Zhang[1‡], Ze You[1‡], Qiaoling Chang[1,3‡], Li Liao** **[1]***

**1** School of Nursing, University of South China, Hengyang, Hunan Province, China, **2** School of Nursing, Yueyang Vocational Technical College, Yueyang, Hunan Province, China, **3** The First Affiliated Hospital of University of South China, Hengyang, Hunan Province, China

⊚ These authors contributed equally to this work.
‡ XH, FZ, ZY and QC also contributed equally to this work.
* 254251558@qq.com

**Data Availability Statement:** No datasets were generated or analysed during the current study. All relevant data from this study will be made available upon study completion.

## Abstract

### Introduction

Several teaching methods have been used in clinical nursing teaching to increase quality and efficiency, but disagreements over their effects persist. This study will evaluate the effects of five teaching methods in clinical nursing on nursing students' knowledge, skill scores, learning satisfaction, and patients' satisfaction.

### Methods

We will conduct searches in PubMed, Embase, Web of Science, The Cochrane Library, China National Knowledge Infrastructure Database (CNKI), China Biological literature database (CBM), Wanfang Database, and China Science and Technology Journal Database (CSTJ) up to April 2022. Relevant randomized controlled trials meeting the eligibility criteria will be included. And the study selection and data extraction will be independently screened for eligibility by two authors. The quality of evidence will be evaluated using the Cochrane risk of bias tool. Pairwise meta-analysis and network meta-analysis (NMA) will be conducted using Rev Man, Stata, and R software. Statistical analyses including homogeneity tests, sensitivity analysis, transitivity tests, consistency tests, and publication bias will be completed.

### Ethics and dissemination

No formal research ethics approval is required. The results will be disseminated to a peer-reviewed journal for publication.

### Protocol registration number

INPLASY2021120040.

**Funding:** This study was funded by Hunan Province Wisdom Ucare Innovation and Entrepreneurship Education Center (Li Liao, No.2018380) and Project of Postgraduate Research Innovation in Hunan Province (Jinhui Ni, No.CX20210918). The funders will not have a role in study design, data collection and analysis, decision to publish, or preparation of the manuscript. No additional external funding was received for this study.

**Competing interests:** The authors have declared that no competing interests exist.

## 1. Introduction

Clinical nursing teaching plays an essential role in the combination of nursing theory and practice [1, 2], helping gradually shape nursing students' attitudes toward learning, practice, and professional development [3]. It provides an authentic context for students to learn how to become registered nurses in knowledge, skills, behaviors, attitudes, and values [4]. The ultimate purpose of clinical practice is to train confident and competent professionals who are dedicated to self-directed learning and patient-centered care [5]. Negative clinical nursing education experiences (such as inadequate acceptance, supervision, and guidance) may decline students' faith in their abilities to practice and lack the preparation to transform into professional nurses [6, 7]. However, with high levels of physical and emotional stress in the COVID-19 pandemic, nurses faced unprecedented challenges and requirements [8, 9]. Nursing students, as a crucial reserve, entail in-person clinical practice in their cultivation, which faces risks and limitations at present, therefore their education should be further upgraded. Meanwhile, operating a high-performing health care delivery system is highly dependent on medical education [10]. Although clinical education is obligatory and valuable, it is not without difficulties and limitations. Therefore, exploring teaching methods is necessary that allow students to understand all the factors in practice-based and complex clinical environments, thus integrating the acquired information into the ability to practice and obtain more experience [11].

Historically, traditional teaching method in which students passively receive knowledge has been used that struggles to meet today's professional needs and standards [12]. Consequently, nurse educators have begun to adopt a variety of teaching methods to increase quality and efficiency in providing clinical nursing teaching [1, 13], such as critical pathways, problem-based learning, patient simulation, case-based learning, mentors, and so on [14]. Although the application of different teaching methods in the clinical environment, disagreements over their effects persist. Indeed, data from several previous meta-analyses have given evidence for the specific teaching methods' efficacy, despite differences in teaching times and outcomes [15–17]. Network meta-analysis (NMAs) offers a way to simultaneously compare interventions and the ranking of interventions based on relative efficacy or safety [18, 19]. To date, no published NMAs combined direct and indirect evidence for these teaching methods in clinical nursing teaching, so there is a critical need to evaluate the effects to make decisions.

This systematic review and network meta-analysis seek to generate conclusive evidence about the effects of five teaching methods on nursing students' knowledge, skill scores, learning satisfaction, and patients' satisfaction, to deliver to nurse educators to guide them in their efforts to improve clinical nursing teaching.

## 2. Materials and methods

This protocol for systematic review and network meta-analysis was conducted under the guidance of the Preferred Reporting Items for Systematic Review and Meta-Analysis Protocols (PRISMA-P) guidelines [20] and the corresponding checklist. The research protocol has been submitted to the INPLASY International Registry of Systematic Reviews, and the number is INPLASY2021120040.

### 2.1. Eligibility criteria

**2.1.1. Types of participants.** Graduate, undergraduate, and junior college nursing students receiving clinical nursing teaching.

**2.1.2. Types of interventions.** One of the following teaching methods used in the experimental group for clinical nursing teaching will be included: critical pathways, problem-based

learning, patient simulation, case-based learning, and mentors. And in which the control group received traditional teaching method.

**2.1.3. Types of outcomes.** The primary outcomes will be objective including nursing students' knowledge and skill scores. And subjective indicators will include nursing students' learning satisfaction and patients' satisfaction with nursing students which are secondary outcomes.

**2.1.4. Types of study.** Randomized controlled studies (RCTs) of teaching methods for nursing students in clinical learning environments.

## 2.2. Exclusion criteria

Excluded studies were as follows: using combined two or more teaching methods in clinical nursing teaching; published articles with incomplete information; duplicate publications; and not published in Chinese or English language.

## 2.3. Data source and search strategy

The following databases will be comprehensively searched: PubMed, Embase, Web of Science, The Cochrane Library, China National Knowledge Infrastructure Database (CNKI), China Biological literature database (CBM), Wanfang Database, and China Science and Technology Journal Database (CSTJ). To avoid omissions, references to included studies and published relevant meta-analyses will be manually searched. And obtain additional research by contacting experts in the field.

The search strategy will be performed according to the PICOS which includes participants (nursing students), interventions (teaching methods), and study types (randomized controlled studies). Through experiments and adjustments, different search strategies adapted for each database will be formed by the combination of medical subject headings (MeSH) and free-text words. The initial search strategy for PubMed is shown in S1 Table.

## 2.4. Study selection

After literature retrieval, NoteExpress software will be used to integrate search results and delete repeated studies. Then, two authors will independently browse the title and abstract of studies to make a preliminary selection. After that, the full text will be read to select the eligible studies. Any discrepancies will be resolved by consensus. The process of study selection is shown in Fig 1.

## 2.5. Data extraction

Two authors will use a data extraction table to extract the data of the included studies. The extracted data will include the following content: basic information (the first author, published journal and year, country, language, etc.), participants' characteristics (sample size of different groups, age, sex, etc.), interventions (random method, intervention types, time, duration, etc.), outcomes and the source of bias. The final dataset will be reviewed to ensure the data was entered correctly.

## 2.6. Risk of bias assessment

According to version 2 of the Cochrane risk-of-bias tool for randomized trials (RoB2) [21], two authors will independently perform the risk of bias analysis of the included study. The tool evaluates the following bias domain: randomization process, deviations from intended interventions, missing outcome data, measurement of the outcome, selection of the reported result, and overall bias. We will grade and score each criterion as lower risk, other, or higher risk of bias. Any discrepancies will be resolved by consensus.

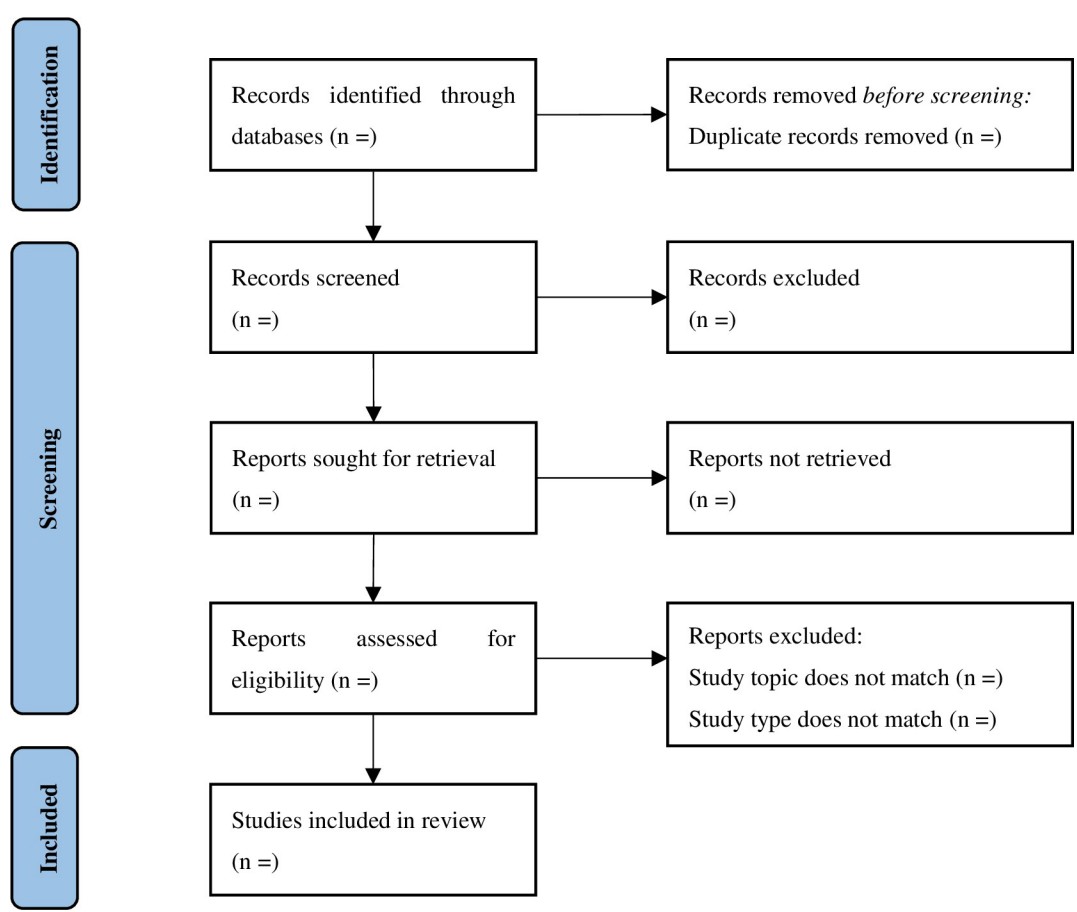

**Fig 1. Flow diagram of study selection.**

## 2.7. Statistical analysis

**2.7.1. Pairwise meta-analysis.** Pairwise meta-analysis will be performed by Rev Man 5.3 when at least three studies assess the same indicator. Given that outcomes in this study are continuous data, effects will be estimated as standardized mean difference (*SMD*) along with 95% confidence intervals (95%*CI*). Heterogeneity will be measured with $I^2$ and *P* values [22]. According to the results, the fixed-effect model ($P<0.10$, $I^2>50\%$) or random-effects model ($P>0.10$, $I^2<50\%$) will be used for analysis. If available, Subgroup analysis will be conducted to explore sources of heterogeneity in pairwise comparison, such as intervention timing and intervention duration. And sensitivity analysis will be carried out if it is necessary, and we will compare results to determine whether lower-quality studies should be excluded.

**2.7.2. Network meta-analysis.** We will select R software to complete the network meta-analysis. The network plot will be drawn by Stata 16.0 to show the direct and indirect comparative relationship between different teaching methods [23]. Heterogeneity test in the same way as pairwise meta-analysis. And a key assumption of NMA is transitivity, it affects the validity of the findings in a network of studies. The decision will be evaluated by comparing study methods and students' characteristics of each study. Afterward, in the case of closed loops of interventions, a consistency test will be conducted by node-splitting analysis [24], and the result will be determined based on the *P* values. The consistency model will be applied ($P>0.05$) when there are no significant differences between direct and indirect comparisons.

Meanwhile, the relative ranking of the different teaching methods will be estimated by the distribution of the ranking probabilities and the surface under the cumulative ranking curves (SUCRA).

**2.7.3. Publication bias.** If more than ten studies are included, funnel plots will be drawn by Rev Man 5.3 and the potential publication bias will be examined through funnel plots.

**2.7.4. Strength of evidence.** Two authors will independently grade the quality of evidence for results with the Grading of Recommendations Assessment, Development, and Evaluation (GRADE) system [25]. The quality of evidence will be classified into four levels: high, moderate, low, and very low.

## 3. Discussion

This systematic review and network meta-analysis is a comprehensive attempt aiming to evaluate five teaching methods in clinical nursing teaching based on direct and indirect evidence of their effect. To determine the effects of these teaching methods, we will rank them according to the advantages and disadvantages of nursing students' knowledge, skill score, and satisfaction. We hope our results may provide nurse educators with better options and new insights into clinical nursing teaching.

## Supporting information

**S1 Table. The initial search strategy for PubMed.**
(DOCX)

**S1 Checklist. This is the PRISMA-P 2015 checklist.**
(DOC)

## Author Contributions

**Conceptualization:** Jinhui Ni, Pei Wu.

**Investigation:** Jinhui Ni, Pei Wu, Xinlin Huang, Fangfang Zhang.

**Methodology:** Jinhui Ni, Xinlin Huang, Fangfang Zhang, Ze You, Qiaoling Chang.

**Resources:** Fangfang Zhang.

**Software:** Ze You.

**Supervision:** Qiaoling Chang, Li Liao.

**Writing – original draft:** Jinhui Ni, Xinlin Huang.

**Writing – review & editing:** Jinhui Ni, Pei Wu, Xinlin Huang, Fangfang Zhang, Ze You, Qiaoling Chang, Li Liao.

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
