## [Decision Letter · Decision Letter 0]

4 Jul 2022

PONE-D-22-11847Effecets of five teaching methods in clinical nursing teaching: A protocol for systematic review and network meta-analysisPLOS ONE

Dear Dr. Liao,

Thank you for submitting your manuscript to PLOS ONE. After careful consideration, we feel that it has merit but does not fully meet PLOS ONE’s publication criteria as it currently stands. Therefore, we invite you to submit a revised version of the manuscript that addresses the points raised during the review process.The authors should critically consider the reviewer's comment regarding the risk of bias and strength of evidence for RCTs.  The search strategy is not extensively addressed the different potential databases for published and gray literatures.They have to also show the geometry of networks to clearly understand the authors intention.Please submit your revised manuscript by Aug 18 2022 11:59PM. If you will need more time than this to complete your revisions, please reply to this message or contact the journal office at plosone@plos.org. Please include the following items when submitting your revised manuscript:A rebuttal letter that responds to each point raised by the academic editor and reviewer(s). You should upload this letter as a separate file labeled 'Response to Reviewers'.A marked-up copy of your manuscript that highlights changes made to the original version. You should upload this as a separate file labeled 'Revised Manuscript with Track Changes'.An unmarked version of your revised paper without tracked changes. You should upload this as a separate file labeled 'Manuscript'.If applicable, we recommend that you deposit your laboratory protocols in protocols.io to enhance the reproducibility of your results. Protocols.io assigns your protocol its own identifier (DOI) so that it can be cited independently in the future. For instructions see: https://journals.plos.org/plosone/s/submission-guidelines#loc-laboratory-protocols. Additionally, PLOS ONE offers an option for publishing peer-reviewed Lab Protocol articles, which describe protocols hosted on protocols.io. Read more information on sharing protocols at https://plos.org/protocols?utm_medium=editorial-email&utm_source=authorletters&utm_campaign=protocols.

We look forward to receiving your revised manuscript.

Kind regards,

Tefera Chane Mekonnen, Master in Public Health(MPH)

Academic Editor

PLOS ONE

Journal Requirements:

2. Please amend either the title on the online submission form (via Edit Submission) or the title in the manuscript so that they are identical.

Reviewers' comments:

Reviewer's Responses to Questions

**Comments to the Author**

1. Does the manuscript provide a valid rationale for the proposed study, with clearly identified and justified research questions?

Reviewer #1: Partly

Reviewer #2: Yes

Reviewer #3: Yes

2. Is the protocol technically sound and planned in a manner that will lead to a meaningful outcome and allow testing the stated hypotheses?

Reviewer #1: Yes

Reviewer #2: Partly

Reviewer #3: Yes

3. Is the methodology feasible and described in sufficient detail to allow the work to be replicable?

Reviewer #1: Yes

Reviewer #2: Yes

Reviewer #3: Yes

4. Have the authors described where all data underlying the findings will be made available when the study is complete?

Reviewer #1: No

Reviewer #2: No

Reviewer #3: Yes

5. Is the manuscript presented in an intelligible fashion and written in standard English?

Reviewer #1: Yes

Reviewer #2: Yes

Reviewer #3: Yes

6. Review Comments to the Author

You may also provide optional suggestions and comments to authors that they might find helpful in planning their study.

Reviewer #1: Abstract:

1. A brief description of how the analysis was carried out must be added.

2. A description of ethics and dissemination of the results must be added as a separate heading under the abstract.

3. Abbreviations must be avoided in Abstract. Eg: VIP and PICOS

4. Keywords can be added.

Introduction:

1. Page 3, Line 42: What do you mean by negative clinical nursing experiences? Clarification is needed.

2. Page 3, Line 44: What are the unprecedented challenges faced by nurses during COVID-19. An explanation can be added before coming to conclusion.

Methods:

1. Page 5, Line 94: You have mentioned excluding English and non-Chinese literature. If you have included literature conducted only in China, kindly add the same in the title.

2. Page 7, Line 120: As per the current version of Cochrane the risk of bias tool is RoB 2 tool. But the reference that you have cited dates back to 2011. Kindly use the recent version of the risk of bias assessment tool for your systematic review. Kindly refer https://training.cochrane.org/handbook/current/chapter-08 for further information.

3. Do you have any specific reason for choosing INPLASY for registering your systematic review? Why did you not consider using PROSPERO?

Reviewer #2: The protocol developed discusses about the usage of systemic review and Network meta-analysis

Network meta-analysis (NMA) studies are used to compare and evaluate the superiority of each intervention. Understanding the concepts and processes of systematic reviews and meta-analyses is essential to understanding NMA. As with systematic reviews and meta-analyses, NMA involves specifying the topic, searching for and selecting all related studies, and extracting data from the selected studies.

To evaluate the effects of each intervention, NMA compares and analyses three or more interventions methods using both direct and indirect evidence.

There is a possibility of several biases when performing NMA. Therefore, key assumptions like similarity, transitivity, and consistency should be satisfied when performing NMA. Among these key assumptions, consistency can be evaluated and quantified by statistical tests. The author has used the Cochrane Collaboration’s tool for assessing risk of bias.

It would be good to add few points on emerging issues in NMA, including methods for evaluation of consistency.

Limitations of the developed protocol is not defined

Reviewer #3: The article entitled Effects of five teaching methods in clinical nursing teaching: A protocol for systematic review and network meta-analysis" is a good piece of work. I have some minor suggestion prior to acceptance;

1. There are some minor language corrections in the article, including typographic and punctuation errors

2. Did the authors omitted scopus? what is the reason?

3. More specific inclusion criteria shoud be mentioned

7. PLOS authors have the option to publish the peer review history of their article (what does this mean?). If published, this will include your full peer review and any attached files.

Reviewer #1: No

Reviewer #2: No

Reviewer #3: No

---

## [Author Response · Author response to Decision Letter 0]

16 Jul 2022

Response to reviewer1:

Abstract:1. A brief description of how the analysis was carried out must be added.

Response: Thank you for your suggestion. We have added the description of the analysis to the abstract.

Revision in the manuscript:

Pairwise meta-analysis and network meta-analysis (NMA) will be conducted using Rev Man, Stata, and R software. Statistical analyses including homogeneity tests, sensitivity analysis, transitivity tests, consistency tests, and publication bias will be completed.

Abstract:2. A description of ethics and dissemination of the results must be added as a separate heading under the abstract. 

Response: Thanks for your suggestion. Ethics and dissemination have been added.

Revision in the manuscript:

Ethics and dissemination: No formal research ethics approval is required. The results will be disseminated to a peer-reviewed journal for publication.

Abstract:3. Abbreviations must be avoided in Abstract. Eg: VIP and PICOS. 

Response: Sorry for the spelling of VIP China Science and Technique Journals Database, which is one of the names of this database. To avoid ambiguity, we have changed it to China Science and Technology Journal Database (CSTJ) after consulting. And abbreviations in the abstract have been revised. 

Revision in the manuscript:

We will conduct searches in PubMed, Embase, Web of Science, The Cochrane Library, China National Knowledge Infrastructure Database (CNKI), China Biological literature database (CBM), Wanfang Database, and China Science and Technology Journal Database (CSTJ) up to April 2022. Relevant randomized controlled trials meeting the eligibility criteria will be included. 

Abstract:4. Keywords can be added. 

Response: Thanks for your suggestion. According to the requirements of journals, keywords (including Teaching Method, Clinical Nursing Teaching, and Network meta-analysis.) have been filled in the submission system, which doesn’t have to be reflected in the manuscript.

Introduction:1. Page 3, Line 42: What do you mean by negative clinical nursing experiences? Clarification is needed. 

Response: Thanks for your questions. Not all clinical settings provide positive learning experiences for nursing students. Negative clinical nursing experiences stem from various influences, such as inadequate supervision by clinical nursing instructors, emotional stressors from the education process, and reluctance by patients to receive care from students. These factors may cause nursing students to have inferior learning experiences, thus affecting the efficiency of nursing education.

Revision in the manuscript:

Negative clinical nursing education experiences (such as inadequate acceptance, supervision, and guidance) may decline students’ faith in their abilities to practice and lack the preparation to transform into professional nurses [6, 7].

Reference:

6. van Rooyen DRM, Jordan PJ, Ten Ham-Baloyi W, Caka EM. A comprehensive literature review of guidelines facilitating transition of newly graduated nurses to professional nurses. Nurse Educ Pract. 2018;30:35-41. https://doi.org/10.1016/j.nepr.2018.02.010 PMID: 29524807

Introduction:2. Page 3, Line 44: What are the unprecedented challenges faced by nurses during COVID-19. An explanation can be added before coming to conclusion. 

Reference: Thanks for your suggestions. We have added the introduction.

Revision in the manuscript:

However, during high levels of physical and emotional stress in the COVID-19 pandemic, there was a certain shortage of clinical nurses [8, 9]. Nursing students, as a crucial reserve, require in-person clinical practice in their cultivation, which faces risks and limitations at present, therefore their education should be further upgraded.

Methods:1. Page 5, Line 94: You have mentioned excluding English and non-Chinese literature. If you have included literature conducted only in China, kindly add the same in the title. 

Response: Thank you for pointing this out. Our included studies will include literature written in Chinese and English, but not limited to China-wide studies. We have revised the methods.

Revision in the manuscript:

2.2 Exclusion criteria

Excluded studies were as follows: using combined two or more teaching methods in clinical nursing teaching; published articles with incomplete information; duplicate publications; and not published in Chinese or English language.

Methods: 2. Page 7, Line 120: As per the current version of Cochrane the risk of bias tool is RoB 2 tool. But the reference that you have cited dates back to 2011. Kindly use the recent version of the risk of bias assessment tool for your systematic review. Kindly refer https://training.cochrane.org/handbook/current/chapter-08 for further information. 

Response: We are very sorry for our negligence. The tool for assessing the risk of bias in randomized trials has been modified to RoB2.

Revision in the manuscript:

2.6 Risk of bias assessment

According to version 2 of the Cochrane risk-of-bias tool for randomized trials (RoB2) [21], two authors will independently perform the risk of bias analysis of the included study. The tool evaluates the following bias domain: randomization process, deviations from intended interventions, missing outcome data, measurement of the outcome, selection of the reported result, and overall bias. We will grade and score each criterion as lower risk, other, or higher risk of bias. Any discrepancies will be resolved by consensus.

Reference

21. Sterne JAC, Savović J, Page MJ, Elbers RG, Blencowe NS, Boutron I, et al. RoB 2: a revised tool for assessing risk of bias in randomised trials. BMJ. 2019;366:l4898. https://doi.org/10.1136/bmj.l4898 PMID: 31462531

Methods:3. Do you have any specific reason for choosing INPLASY for registering your systematic review? Why did you not consider using PROSPERO? 

Response: Thanks for your questions. This systematic review is related to academic requirements as the author is a graduate student. The main reason for choosing INPLASY is that it provides a faster way to register than PROSPERO. When meeting requirements, it will be published in less than 48h.

Response to reviewer2:

It would be good to add few points on emerging issues in NMA, including methods for evaluation of consistency. Limitations of the developed protocol is not defined. 

Response: Thank you for your suggestion. And your interpretation of NMA had benefited us a lot. We have added some information about it.

Revision in the manuscript:

2.7.2 Network meta-analysis

We will select R software to complete the network meta-analysis. The network plot will be drawn by Stata 16.0 to show the direct and indirect comparative relationship between different teaching methods [23]. Heterogeneity test in the same way as pairwise meta-analysis. And a key assumption of NMA is transitivity, it affects the validity of the findings in a network of studies. The decision will be evaluated by comparing study methods and students’ characteristics of each study. Afterward, in the case of closed loops of interventions, a consistency test will be conducted by node-splitting analysis [24], and the result will be determined based on the P values. The consistency model will be applied (P＞0.05) when there are no significant differences between direct and indirect comparisons. Meanwhile, the relative ranking of the different teaching methods will be estimated by the distribution of the ranking probabilities and the surface under the cumulative ranking curves (SUCRA).

Response to reviewer3:

1. There are some minor language corrections in the article, including typographic and punctuation errors 

Response: We are very sorry for these kinds of mistakes. They should not happen. We have done the correction for all identified errors in the manuscript.

2. Did the authors omitted scopus? what is the reason? 

Response: Thanks for your question. In China, Scopus is not a commonly used foreign language database in nursing education. Referring to the relevant literature, English databases frequently select PubMed, Embase, Web of Science, The Cochrane Library, and CINAHL. And we made some adjustments in this part.

Revision in the manuscript:

2.3 Data source and search strategy

The following databases will be comprehensively searched: PubMed, Embase, Web of Science, The Cochrane Library, China National Knowledge Infrastructure Database (CNKI), China Biological literature database (CBM), Wanfang Database, and China Science and Technology Journal Database (CSTJ). To avoid omissions, references to included studies and published relevant meta-analyses will be manually searched. And obtain additional research by contacting experts in the field.

The search strategy will be performed according to the PICOS which includes participants (nursing students), interventions (teaching methods), and study types (randomized controlled studies). Through experiments and adjustments, different search strategies adapted for each database will be formed by the combination of medical subject headings (MeSH) and free-text words. The initial search strategy for PubMed is shown in Table 1.

3. More specific inclusion criteria should be mentioned. 

Response: Thank you for your suggestion. We have added more details about it. 

Revision in the manuscript:

2.1 Eligibility criteria

2.1.1 Types of participants 

Graduate, undergraduate, and junior college nursing students receiving clinical nursing teaching.

2.1.2 Types of interventions

One of the following teaching methods used in the experimental group for clinical nursing teaching will be included: critical pathways, problem-based learning, patient simulation, case-based learning, and mentors. And in which the control group received traditional teaching method.

2.1.3 Types of outcomes

The primary outcomes will be objective including nursing students' knowledge and skill scores. And subjective indicators will include nursing students' learning satisfaction and patients' satisfaction with nursing students which are secondary outcomes.

2.1.4 Types of study

Randomized controlled studies (RCTs) of teaching methods for nursing students in clinical learning environments.

---

## [Editor Report · Decision Letter 1]

15 Aug 2022

Effects of five teaching methods in clinical nursing teaching: A protocol for systematic review and network meta-analysis

PONE-D-22-11847R1

Dear Dr. Liao,

We’re pleased to inform you that your manuscript has been judged scientifically suitable for publication and will be formally accepted for publication once it meets all outstanding technical requirements.

Kind regards,

Tefera Chane Mekonnen, Master in Public Health(MPH)

Academic Editor

PLOS ONE
---

## [Editor Report · Acceptance letter]

19 Aug 2022

PONE-D-22-11847R1 

Effects of five teaching methods in clinical nursing teaching: A protocol for systematic review and network meta-analysis 

Dear Dr. Liao:

I'm pleased to inform you that your manuscript has been deemed suitable for publication in PLOS ONE. Congratulations! Your manuscript is now with our production department. 

Kind regards, 

on behalf of

Dr. Tefera Chane Mekonnen 

Academic Editor

PLOS ONE